# Are there any 'object detectors' in the hidden layers of CNNs trained to identify objects or scenes?

## Abstract

Various methods of measuring unit selectivity have been developed with the aim of better understanding how neural networks work. But the different measures provide divergent estimates of selectivity, and this has led to different conclusions regarding the conditions in which selective object representations are learned and the functional relevance of these representations. In an attempt to better characterize object selectivity, we undertake a comparison of various selectivity measures on a large set of units in AlexNet, including localist selectivity (Bowers et al., 2014), precision (Zhou et al., 2015), class-conditional mean activity selectivity (CCMAS) (Morcos et al., 2018), network dissection (Zhou et al., 2018a), the human interpretation of activation maximization (AM) images, and standard signal-detection measures. We find that the different measures provide different estimates of object selectivity, with precision and CCMAS measures providing misleadingly high estimates. Indeed, the most selective units had a poor hit-rate or a high false-alarm rate (or both) in object classification, making them poor object detectors. We fail to find any units that are even remotely as selective as the 'grandmother cell' units reported in recurrent neural networks. In order to generalize these results, we compared selectivity measures on a few units in VGG-16 and GoogLeNet trained on the ImageNet or Places-365 datasets that have been described as 'object detectors'. Again, we find poor hit-rates and high false-alarm rates for object classification.

## 1 Introduction

There have been recent attempts to understand how neural networks (NNs) work by analyzing hidden units one-at-a-time using various measures such as localist selectivity (Bowers et al., 2014), class-conditional mean activity selectivity (CCMAS) (Morcos et al., 2018), precision (Zhou et al., 2015), network dissection (Zhou et al., 2018a), and activation maximization (AM) (Erhan et al., 2009). These measures are all taken to provide evidence that some units respond highly selectively to categories of objects under some conditions. Not only are these findings surprising given the widespread assumption that NNs only learn highly distributed and entangled representations, they raise a host of questions, including the functional importance of these selective representations (Zhou et al., 2018b), the conditions in which they are learned (e.g., Morcos et al., 2018), and the relation between these representations and the selective neurons observed in cortex (Bowers, 2009).

To answer these question, it is necessary to have a better understanding of what these metrics actually measure, and how they relate to one another. Accordingly, we directly compare these measures of selectivity on the same set of units as well as adopt standard signal-detection measures in an attempt to provide better measures of single-unit selectivity to object category. In addition, to provide a more intuitive assessment of selectivity, we report jitterplots for a few of the most selective units that visually display how the unit responds to the different image categories. We focus on AlexNet (Krizhevsky et al., 2012) trained on ImageNet (Deng et al., 2009) because many authors have studied the selectivity of single hidden units in this model using a range of quantitative (Zhou et al., 2018a; 2015) and qualitative (Nguyen et al., 2017; Yosinski et al., 2015; Simonyan et al., 2013) methods. But we also compare different selectivity measures on specific units in VGG-16 (Simonyan and Zisserman, 2014) and GoogLeNet (Szegedy et al., 2015) trained on the the ImageNet and Places-365

datasets that were characterized by Zhou et al. (2018a) as "object detectors" based on their Network Dissection method (Zhou et al., 2018a). Our main findings are:

1. The precision and CCMAS measures are misleading with near-maximum selectivity scores associated with units that strongly respond to many different image categories. By contrast, the signal-detection measures more closely capture the level of selectivity displayed in the jitterplots (Sec. 3.1).

2. Units with interpretable AM images do not correspond to highly selective representations (Sec. 3.2).

3. The Network Dissection method also provides a misleading measure for "object detectors" (Sec. 3.3).

In one line of research, Bowers et al. (2014; 2016) assessed the selectivity of single hidden units in recurrent neural networks (RNNs) designed to model human short-term memory. They reported many 'localist' or 'grandmother cell' units that were 100% selective for specific letters or words, where all members of the selective category were more active than and disjoint from all non-members, as can be shown in jitterplots (Berkeley et al., 1995) (see Fig. 1 for a unit selective to the letter 'j'). The authors argued that the network learned these representations in order to co-activate multiple letters or words at the same time in short-term memory without producing ambiguous blends of overlapping distributed patterns (the so-called 'superposition catastrophe'). Consistent with this hypothesis, localist units did not emerge when the model was trained on letters or words one-at-a-time (Bowers et al., 2014) (see Fig. 1 for an example of a non-selective unit).

In parallel, researchers have reported selective units in the hidden layers of various CNNs trained to classify images into one of multiple categories (Zhou et al., 2015; Morcos et al., 2018; Zeiler and Fergus, 2014; Erhan et al., 2009), for a review see Bowers (2017). For example, Zhou et al. (2015) assessed the selectivity of units in the pool5 layer of two CNNs trained to classify images into 1000 objects and 205 scene categories, respectively. They reported many highly selective units that they characterized as 'object detectors' in both networks. Similarly, Morcos et al. (2018) reported that CNNs trained on CIFAR-10 and ImageNet learned many highly selective hidden units, with CCMAS scores approaching the maximum of 1.0. These later findings appear to be inconsistent with Bowers et al. (2016) who failed to observe selective representations in fully connected NNs trained on stimuli one-at-a-time (see Fig. 1), but the measures of selectivity that have been applied across studies are different, and accordingly, it is difficult to directly compare results.

A better understanding of the relation between selectivity measures is vital given that different measures are frequently used to address similar issues. For example, both the human interpretability of generated images (Le, 2013) and localist selectivity (Bowers et al., 2014) have been used to make claims about 'grandmother cells', but it is not clear whether they provide similar insights into unit selectivity. Similarly, based on their precision metric, Zhou et al. (2015) claim that the object detectors learned in CNNs play an important role in identifying specific objects, whereas Morcos et al. (2018) challenge this conclusion based on their finding that units with high CCMAS measures were not especially important in the performance of their CNNs and concluded: "...it implies that methods for understanding neural networks based on analyzing highly selective single units, or finding optimal inputs for single units, such as activation maximization (Erhan et al., 2009) may be misleading". This makes a direct comparison between selectivity measures all the more important.

In order to directly compare and have a better understanding of the different selectivity measures we assessed (1) localist, (2) precision, and (3) CCMAS selectivity of the conv5, fc6, and fc7 of AlexNet trained on ImageNet, and in addition, we employed a range of signal detection methods on these units, namely, (4) recall with 100% and 95% precision, (5) maximum informedness, (6) specificity at maximum informedness , and (7) recall (also called *sensitivity*) at maximum informedness, and false alarm rates at maximum informedness (described in Sec. 2). We also assessed the selectivity of a few units in VGG-16 and GoogLeNet models trained on the ImageNet and Places-365 dataset that were highly selective according to the Network Dissection method (Zhou et al., 2018a). We show that the precision and CCMAS measures often provide misleadingly high estimates of object selectivity compared to other measures, and we do not find any units that can be reasonably described as 'object detectors' given that the most selective units show a low hit-rate or a high false-alarm rate (or both) when classifying images. At best, the most selective units in CNNs are sensitive to some unknown feature that is weakly associated with the class in question.

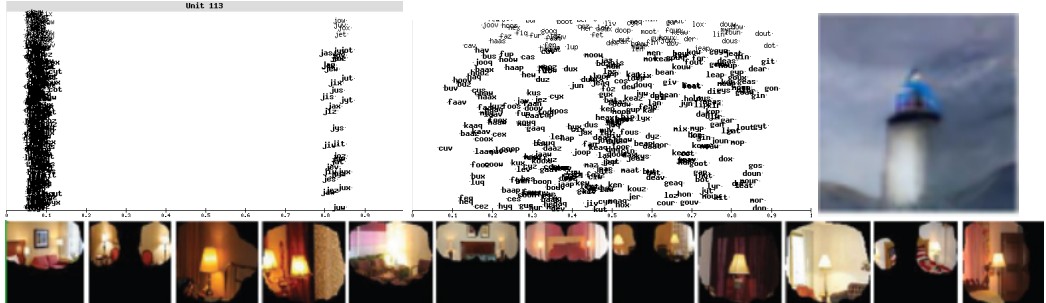

Figure 1: Examples of selectivity measures used. **Top left:** jitterplot of unit 113 in an RNN (under the superposition constraint) selective to the letter 'j' (Bowers et al., 2016). **Top middle:** jitterplot of a non-selective unit 160 found in an RNN trained on words one-at-a-time from (Bowers et al., 2016). **Top right:** Activation maximization image of unit conv5$_9$ AlexNet that resembles a lighthouse (Nguyen et al., 2016). **Bottom:** highest-activation images for a 'lamp' detector with 84% precision in the layer conv5 of AlexNet; from (Zhou et al., 2015).

In addition to these quantitative measures and jitterplots we assessed selectivity with a common qualitative measure, namely, human interpretation of images generated by a state-of-the-art activation maximization (AM) method (Nguyen et al., 2017). AM images are generated to strongly activate individual units, and some of them are interpretable by humans (e.g., a generated image that looks like a lighthouse, see Fig. 1). For the first time, we systematically evaluated the interpretability of the AM images and compare these ratings with the selectivity measures for corresponding units. We show that the few hidden units with interpretable AM images are not highly selective.

## 2 METHODS

**Network and Dataset** All ∼1.3M photos from the ImageNet ILSVRC 2012 dataset (Deng et al., 2009) were cropped to $277 \times 277$ pixels and classified by the pre-trained AlexNet CNN (Krizhevsky et al., 2012) shipped with Caffe (Jia et al., 2014), resulting in 721,536 correctly classified images. Once classified, the images are not re-cropped nor subject to any changes. We analyzed the fully connected (fc) layers: fc6 and fc7 (4096 units), and the top convolutional layer conv5 which has 256 filters. We only recorded the activations of correctly classified images. The activation files are stored in .h5 format and will be available at `http://anonymizedForReview`. We randomly selected 233 conv5, 2738 fc6, 2239 fc7 units for analysis.

**Localist selectivity** Following Bowers et al. (2014), we define a unit to be localist for class $A$ if the set of activations for class $A$ was higher and disjoint with those of $\neg A$. Localist selectivity is easily depicted with jitterplots (Berkeley et al., 1995) in which a scatter plot for each unit is generated (see Figs. 1 and 3). Each point in a plot corresponds to a unit's activation in response to a single image, and only correctly classified images are plotted. The level of activation is coded along the $x$-axis, and an arbitrary value is assigned to each point on the $y$-axis.

**Precision** Precision refers to the proportion of items above some threshold from a given class. The precision method of finding object detectors involves identifying a small subset of images that most strongly activate a unit and then identifying the critical part of these images that are responsible for driving the unit. Zhou et al. (2015) took the 60 images that activated a unit the most strongly and asked independent raters to interpret the critical image patches (e.g., if 50 of the 60 images were labeled as 'lamp', the unit would have a precision index of 50/60 or 83%; see Fig. 1). Object detectors were defined as units with a precision score $> 75\%$: they reported multiple such detectors. Here, we approximate this approach by considering the 60 images that most strongly activate a given unit and assess the highest percentage of images from a given output class.

**CCMAS** Morcos et al. (2018) introduced a selectivity index called the Class-conditional Mean Activation Selectivity (CCMAS). The CCMAS for class $A$ compares the mean activation of all images in class $A$, $\mu_A$, with the mean activation of all images not in class $A$, $\mu_{\neg A}$, and is given by: $(\mu_A - \mu_{\neg A}) / (\mu_A + \mu_{\neg A})$. Here, we assessed class selectivity for the highest mean activation class.

**Activation Maximization** We harnessed an activation maximization method called Plug & Play Generative Networks (Nguyen et al., 2017) in which an image generator network was used to generate images (AM images) that highly activate a unit in a target network. We used the public code released by Nguyen et al. (2017) and their default hyperparameters.[1] We generated 100 separate images that maximally activated each unit in the conv5, fc6, and fc8 layers of AlexNet and asked participants to judge whether they could identify any repeating objects, animals, or places in images in a behavioral experiment (Sec. 3.2). Readers can test themselves at: `https://research.sc/participant/login/dynamic/63907FB2-3CB9-45A9-B4AC-EFFD4C4A95D5`

**Recall with perfect and 95% precision** Recall with perfect and 95% precision are related to localist selectivity except that they provide a continuous rather than discrete measure. For recall with perfect precision we identified the image that activated a given unit the most and counted the number of images from the same class that were more active than all images from all other classes. We then divided this result by the total number of correctly identified images from this class. A recall with a perfect precision score of 1 is equivalent to a localist representation. Recall with a 95% precision allows 5% false alarms.

**Maximum informedness** Maximum informedness identifies the class and threshold where the highest proportion of images above the threshold and the lowest proportion of images below the threshold are from that class (Powers, 2011). The informedness is computed for each class at each threshold, with the highest value selected. Informedness summarises the diagnostic performance of unit for a given class at a certain threshold based on the recall [True Positives / (True Positives + False Negatives)] and specificity [True Negatives / (True Negatives + False Positives)] in the formula [informedness = recall + specificity − 1] (Powers, 2011).

**Sensitivity or Recall at Maximum Informedness** For the threshold and class selected by Maximum Informedness, recall (or hit-rate) is the proportion of items from the given class that are above the threshold. Also known as true postive rate.

**Specificity at Maximum Informedness** For the threshold and class selected by Maximum Informedness, the proportion of items that are not from the given class that are below the threshold. Also known as true negative rate.

**False Alarm Rate at Maximum Informedness** For the threshold and class selected by Maximum Informedness, the proportion of items that are not from the given class that are above the threshold.

**Network Dissection** To assess the selectivity of a unit in the Network Dissection technique, Zhou et al. (2018a) compute the Intersection over Union (IoU) of an annotated input image $L_c$, for the set of all 'concepts' $c$ and a spatial activation map, $M_k$, of where a unit $k$ is. A unit $k$ is taken as a detector for concept $c$ if its $IoU_{k,c}$ exceeds a pre-defined threshold $T$. See Zhou et al. (2018a) for more details.

## 3 RESULTS

### 3.1 COMPARISON OF SELECTIVITY MEASURES IN ALEXNET

The results from the various of selectivity measures applied to the conv5, fc6, and fc7 layers of AlexNet are displayed in Fig. 2a–i. We did not plot the localist selectivity as there were no localist 'grandmother units'. The first point to note is that multiple units in the fc6 and fc7 layers had near 100% precision scores and multiple units had CCMAS scores approaching 1. For example, in layer fc7, we found 14 units with a precision > 0.9, and 1487 units with a CCMAS > 0.9. The second point is that other measures provided much reduced estimates of selectivity. For example, the unit with the highest recall with a perfect precision score was only .08 (unit 255 responding to images of Monarch butterflies), and the unit with the top maximum informedness score (unit 3290 also responding to images of Monarch butterflies with a score of 0.91) had a false alarm rate above its optimal threshold > 99% (indeed the minimum false alarm rate was 0.96).

To illustrate the contrasting measures of selectivity consider unit $fc6_{1199}$ depicted in Fig. 3 that has a precision score of 98% and a CCMAS score of .92. By Zhou et al.'s criterion, this is a 'Monarch Butterfly' detector (its precision score > 75%). By contrast, the scatter plot and signal-detection

---

[1] `https://github.com/Evolving-AI-Lab/ppgn`

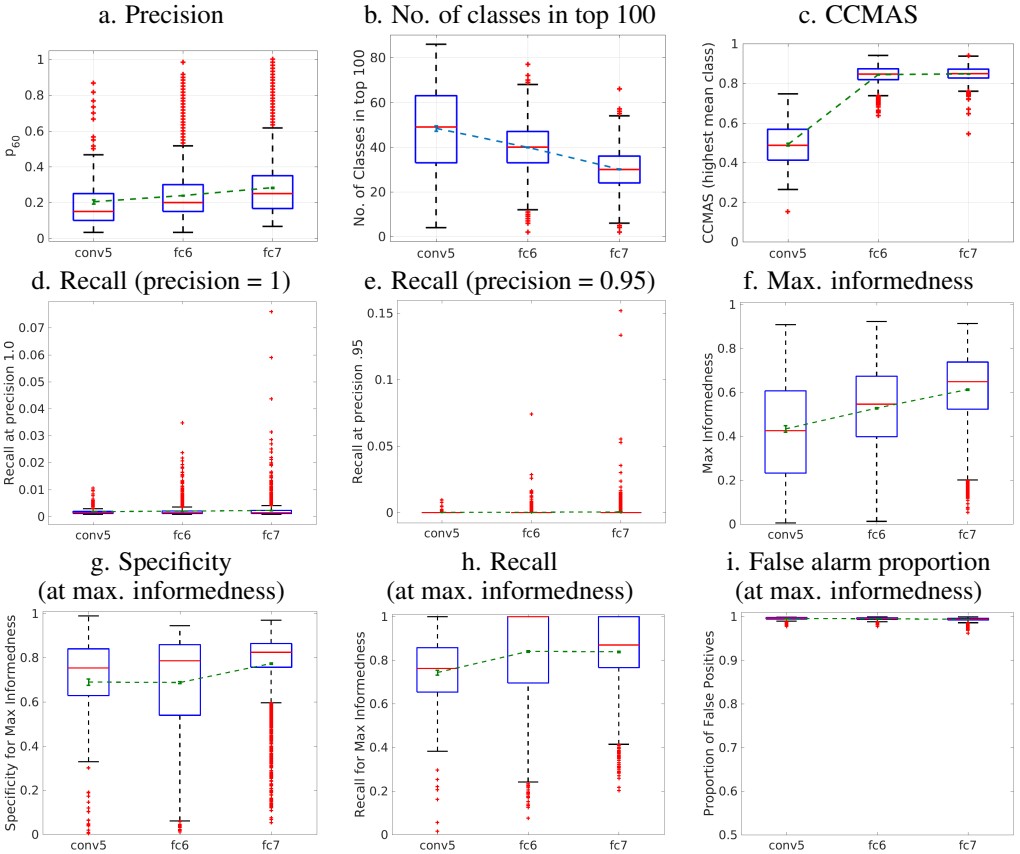

Figure 2: Different selectivity measures across the conv5, fc6, and fc7 layers of AlexNet. Red-line: median of data, top and bottom of box edges is the $25^{\text{th}}$ and $75^{\text{th}}$ percentile, whiskers extend to extreme edges of distribution not considered outliers and red crosses are outliers. Green points and dashed lines are the means of the distributions with standard errors. The high levels of selectivity observed with the precision and CCMAS measures are in stark contrast with the low levels of selectivity observed with the recall with perfect precision and high false-alarm rates at maximum informedness.

scores show this is a mischaracterisation of this unit given that the false alarm rate at maximum informedness was greater than 99% and the modal response to Monarch butterflies was zero.

What level of selectivity is required before a unit can be considered an 'object detector' for a given category? In the end, this is a terminological point. On an extreme view, one might limit the term to the 'grandmother units' that categorize objects with perfect recall and specificity, or alternatively, it might seem reasonable to describe a unit as a detector for a specific object category if there is some threshold of activation that supports more hits than misses (the unit is strongly activated by the majority of images from a given category), and at the same time, supports more hits than false alarms (the unit is strongly activated by items from the given category more often than by items from other categories). Or perhaps a lower standard could be defended, but in our view, the term "object detector" suggests a higher level of selectivity than 8% recall at perfect precision. That said, our results show that some units respond strongly to some (unknown) features that are weakly correlated with an object category. For instance, unit $\text{fc6}_{1199}$ is responding to features that occur more frequently in Monarch Butterflies than other categories. This can also be seen in a recent ablation study in which removing the most selective units tended to impair the CNN's performance in identifying the corresponding object categories more than other categories (Zhou et al., 2018b). But again, the pattern of performance is not consistent with the units being labeled 'object detectors'.

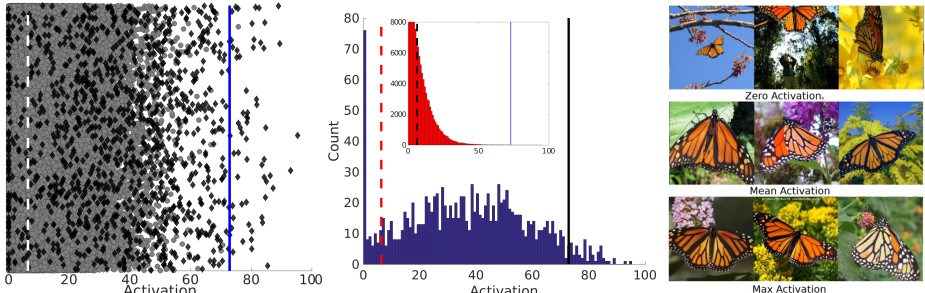

Figure 3: Data for unit fc6$_{1199}$. **Left:** activation jitterplot, black diamonds: Monarch butterfly images; grey circles: all other classes; white dashed line: threshold for the butterfly class maximum informedness; blue solid line: threshold for top 60 activations. **Middle:** histogram of activations of Monarch butterflies; red dashed line: threshold for the butterfly class maximum informedness; black solid line: threshold for top 60 activations. **Inset:** zoomed-in histogram of all activations across all ImageNet classes of unit fc6$_{1199}$ (N.B. this plot shows only the highest 121,586 activations; there are 596,734 activations at 0). There are Monarch butterfly images covering the whole range of values, with 72 images (5.8% of the total) having an activation of 0. **Right:** example ImageNet images with activations of 0 (top), the mean, 39.2±0.6, (middle), and the maximum, 95, (bottom) of the range. Although the high precision score suggests that this unit is a butterfly detector this is misleading given there are butterfly images over the entire activation range (including 0).

## 3.2 HUMAN INTERPRETATION OF ACTIVATION MAXIMIZATION IMAGES FOR ALEXNET UNITS

Activation Maximization is one of the most commonly used interpretability methods for explaining what a single unit has learned in many artificial CNNs and even biological neural networks (see Nguyen et al. (2019) for a survey). Our behavioral experiment provides the first quantitative assessment of AM images and compares AM interpretability to other selectivity measures.

Table 1: Human judgements of whether AM images look like familiar objects in layers conv5, fc6, and fc8 in AlexNet.

| layer | % 'yes' responses | % units ≥ 80% 'yes' response | % overlap between humans and: | | |
|---|---|---|---|---|---|
| | | | humans | most active object | CCMAS class |
| | (a) | (b) | (c) | (d) | (e) |
| conv5 | 21.7% (±1.1%) | 4.3% (± 1.3%) | 89.5% (±5.7% ) | 34.1% (±14.4%) | 0% |
| fc6 | 21.0% (±0.4%) | 3.1% (± 0.4%) | 80.4% (±4.1%) | 23.3% (±5.9%) | 18.9% (±5.9%) |
| fc8 | 71.2% (±0.6%) | 59.3% (±1.6%) | 96.5% (±0.4%) | 95.4% (±0.6%) | 94.6% (±0.7%) |

We generated 100 AM images images for every unit in the layers conv5, fc6, and fc8 in AlexNet, as in Nguyen et al. (2017), and displayed them as $10 \times 10$-image panels. A total of 3,299 image panels were used in the experiment (995 fc8, 256 conv5, and 2048 randomly selected fc6 image panels) and were divided into 64 counterbalanced lists for testing. To assess the interpretability for these units as object detectors, 333 paid volunteers were asked to look at image panels and asked if the images had an object / animal or place in common. If the answer was 'yes', they were asked to write down a generic name for that object (e.g. "fish" rather than "goldfish"). Analyses of common responses was done for any units where over 80% of humans agreed there was an object present.

The results are summarized in Table 1. Not surprisingly, the AM images for output fc8 units are the most human-recognizable as objects across the AlexNet layers (71.2%; Table 1a). In addition, when they were given a consistent interpretation, they almost always (95.4%; Table 1d) match the corresponding ImageNet category. By contrast, less than 5% of units in conv5 or fc6 were associated with consistently interpretable images (Table 1b), and the interpretations only weakly matched the category associated with the highest-activation images or CCMAS selectivity (Table 1d–e). Apart from showing that there are few interpretable units in the hidden layers of AlexNet, our findings show that the interpretability of images does not imply a high level of selectivity given the signal-detection

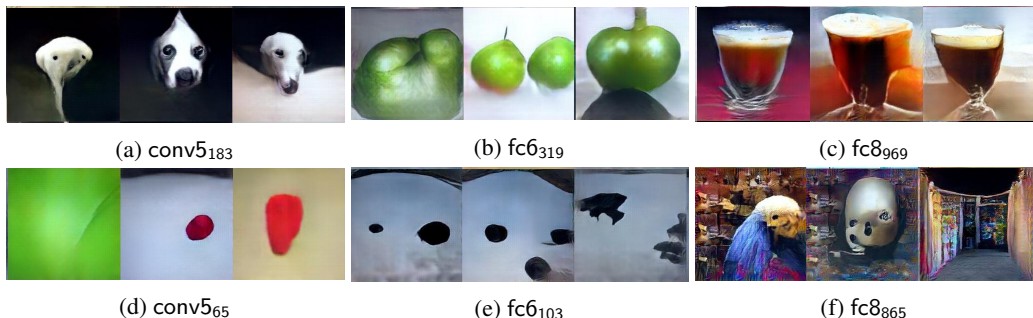

(a) conv5$_{183}$        (b) fc6$_{319}$        (c) fc8$_{969}$

(d) conv5$_{65}$        (e) fc6$_{103}$        (f) fc8$_{865}$

Figure 4: Example AM images that were either judged by all participants to contain objects (a–c) or to be uninterpretable as objects (d–f). The human label for unit conv5$_{183}$ (a) was 'dogs'; the most active image was of a 'flat-coated retriever'; CCMAS class was 'monitor'. For fc6$_{319}$ (b), subjects reported 'green peppers' or 'apples' (all classified as the same broad class in our analysis); both the most active item and CCMAS class were 'Granny Smith apples'. For fc8$_{969}$ (c), humans suggested 'beverage' or 'drink'; both the most active item and CCMAS class were 'eggnog'.

results (Fig. 2d–h). See Fig. 4 for an example of the types of images that participants rated as objects or non-objects.

### 3.3 COMPARING SELECTIVITY MEASURES IN OTHER CNNS

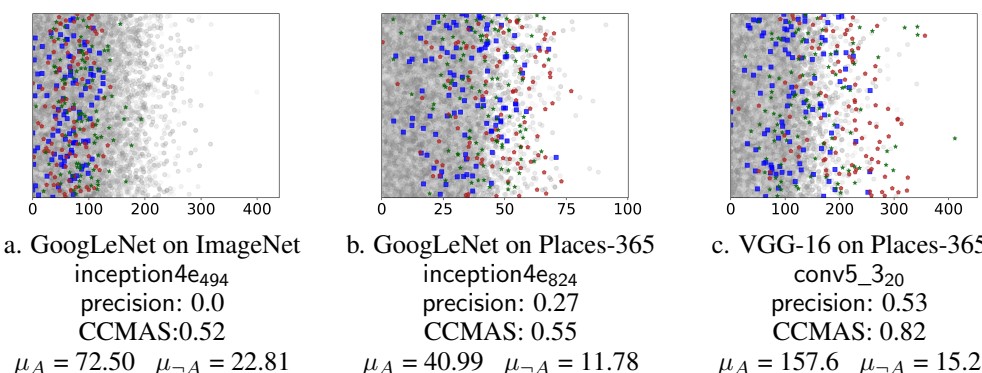

a. GoogLeNet on ImageNet
inception4e$_{494}$
precision: 0.0
CCMAS:0.52
$\mu_A = 72.50$   $\mu_{\neg A} = 22.81$

b. GoogLeNet on Places-365
inception4e$_{824}$
precision: 0.27
CCMAS: 0.55
$\mu_A = 40.99$   $\mu_{\neg A} = 11.78$

c. VGG-16 on Places-365
conv5_3$_{20}$
precision: 0.53
CCMAS: 0.82
$\mu_A = 157.6$   $\mu_{\neg A} = 15.2$

Figure 5: The units with with the highest Network Dissection scores for the category 'bus'. The scatter plots, precision, and CCMAS scores all indicate a low selectivity for this category. **blue** squares: 'school bus'; **red** pentagons: 'trolleybus'; **green** stars: 'minibus'; **grey** circles: other classes.

Thus far we have assessed the selectivity of hidden units in AlexNet and shown that no units can reasonably be characterized as object detectors despite the high precision and CCMAS scores of some units. This raises the question as to whether more recent CNNs learn object detector units. In order to address this we display jitterplots for three units that have the highest IoU scores according to the Network Dissection for the category BUS in (a) GoogLeNet trained on ImageNet, (b) GoogLeNet trained on Places-365, and (c) VGG-16 trained on Places-365, respectively (Zhou et al., 2018a). Models trained on the Places-365 dataset learn to categorize images into scenes (e.g., bedrooms, kitchens, etc.) rather than into object categories, and nevertheless, Zhou et al. (2018a) reported more object detectors in the former models. We illustrate the selectivity of the BUS category because it is an output category in ImageNet so we can easily plot the jitterplots for these units.

As was the case with AlexNet, the jitterplots show that the most selective units show some degree of selectivity, with the BUS images more active on average compared to non-Buses, and the percentage of nonzero activations for BUS higher than the non-BUS categories (see tables A3 - A5 in the appendix for summary of more units). But the units are no more selective than the units we observed in AlexNet. Indeed, the precision measure of selectivity for the first units is 0.0, with none of the

units having a precision of .75 that was the criterion of object detectors by Zhou et al. (2015), and CCMAS scores for first two units were roughly similar to the mean CCMAS score for AlexNet units in conv 5 (and much lower than the mean in fc6 and fc7). The most selective VGG-16 unit trained on Places-365 has lower precision and CCMAS scores than the Monarch Butterfly unit depicted in Figure 3. So again, different measures of selectivity provide support different conclusions, and even the most selective units are far from the selective units observed in recurrent networks as reported in Figure 1a. See tables A3 - A5 in the appendix for more details about these three units.

## 4 DISCUSSIONS AND CONCLUSIONS

Our central finding is that different measures of single-unit selectivity for objects support very different conclusions when applied to the same units in AlexNet. In contrast with the precision (Zhou et al., 2015) and CCMAS (Morcos et al., 2018) measures that suggest some highly selective units for objects in layers conv5, fc6, and fc7, the recall with perfect precision and false alarm rates at maximum informedness show low levels of selectivity. Indeed, the most selective units have a poor hit-rate or a high false-alarm rate (or both) for identifying an object class. The same outcome was observed with units in VGG-16 and GoogLeNet trained on either ImageNet or the Places-365 dataset.

Not only do the different measures provide very different assessments of selectivity, the precision, CCMAS, and Network Dissection measures provide highly misleading estimates of selectivity that have led to mistaken conclusions. For example, unit $fc6_{1199}$ in AlexNet trained on ImageNet is considered an Monarch Butterfly detector according to Zhou et al. (2015) with a precision score of 98% (and a CCMAS score of .93). But the jitterplot in Fig. 3 and signal detection scores (e.g., high false alarm rate at maximum informedness) show this is a mischaracterisation of this unit. In the same way, the Network Dissection method identified many object detectors in VGG-16 and GoogLeNet CNNs, but the jitterplots in Fig. 5 (and precision scores) show that this conclusion is unjustified. For additional problems with the CCMAS score see Figure 5 in Appendix C. Similarly, the images generated by Activation Maximization also provided a misleading estimate of selectivity given that interpretable images were associated with very low selectivity scores. This has led to confusions that have delayed theoretical progress. For example, describing single units in CNNs as "object detectors" in response to high precision measures (Zhou et al.) suggests similar types of representations are learned in CNNs and RNNs. Indeed, we are not aware of anyone in the machine learning community who has even considered the hypothesis that selectivity is reduced in CNNs compared RNNs. Our findings highlight the contrasting results.

What should be made of the finding that localist representations are sometimes learned in RNNs (units with perfect specificity and recall), but not in AlexNet and related CNNs? The failure to observe localist units in the hidden layers of these CNNs is consistent with Bowers et al. (2014)'s claim that these units emerge in order to support the co-activation of multiple items at the same time in short-term memory. That is, localist representations may be the solution to the superposition catastrophe, and these CNNs only have to identify one image at a time. The pressure to learn highly selective representations in response to the superposition constraint may help explain the reports of highly selective neurons in cortex given that the cortex needs to co-activate multiple items at the same time in order to support short-term memory (Bowers et al., 2016).

Note, the RNNs that learned localist units were very small in scale compared to CNNs we have studied here, and accordingly, it is possible that the contrasting results reflect the size of the networks rather than the superposition catastrophe *per se*. Relevant to this issue a number of authors have reported the existence of selective units in larger RNNs with long-short term memory (LSTM) units (Karpathy et al., 2016; Radford et al., 2017; Lakretz et al., 2019; Na et al., 2019). Indeed, Lakretz et al. (2019) use the term 'grandmother cell' to describe the units they observed. It will be interesting to apply our measures of selectivity to these larger RNNs and see whether these units are indeed 'grandmother units'.

It should also be noted that there are recent reports of impressively selective representations in Generative Adversarial Networks (Bau et al., 2019) and Variational Autoencoders (Burgess et al., 2018) where the superposition catastrophe is not an issue. Again, it will be interesting to assess the selectivity of these units according to signal detection measures in order to see whether there are additional computational pressures to learn highly selective or even grandmother cells. We will be exploring these issues in future work.

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

APPENDIX

## A    METHODOLOGICAL DETAILS FOR THE BEHAVIORAL EXPERIMENT

One hundred generated images were made for every unit in layers conv5, fc6 and fc8 in AlexNet, as in Nguyen et al. (2017), and displayed as 10x10 image panels (figures A4 and Figures A2 and A3). A total of 3,299 image panels were used in the experiment (995 fc8, 256 conv5, and 2048 randomly selected fc6 image panels) and were divided into 64 counterbalanced lists of 51 or 52 (4 conv5, 15 or 16 fc8 and 32 fc6). 51 of the lists were assigned to 5 participants and 13 lists were assigned to 6 participants.

To test the interpretability of these units, paid volunteers were asked to look at image panels and asked if the images had an object / animal or place in common. If the answer was 'yes', they were asked to name that object simply (i.e. fish rather than goldfish). Analyses of common responses was carried out for any units where over 80% of humans agreed there was an object present, by reading the human responses and comparing them to both each other and to the output classes. Agreement was taken if the object was the same rough class. For example, 'beer', 'glass', and 'drink' were all considered to be in agreement in the general object of 'drink', and in agreement with both the classes of 'wine glass' and 'beer' as these classes were also general drink classes (this is an actual example, most responses were more obvious and required far less interpretation than that). Participants were given six practice trials, each with panels of 20 images before starting the main experiment. Practice trials included images that varied in their interpretability.

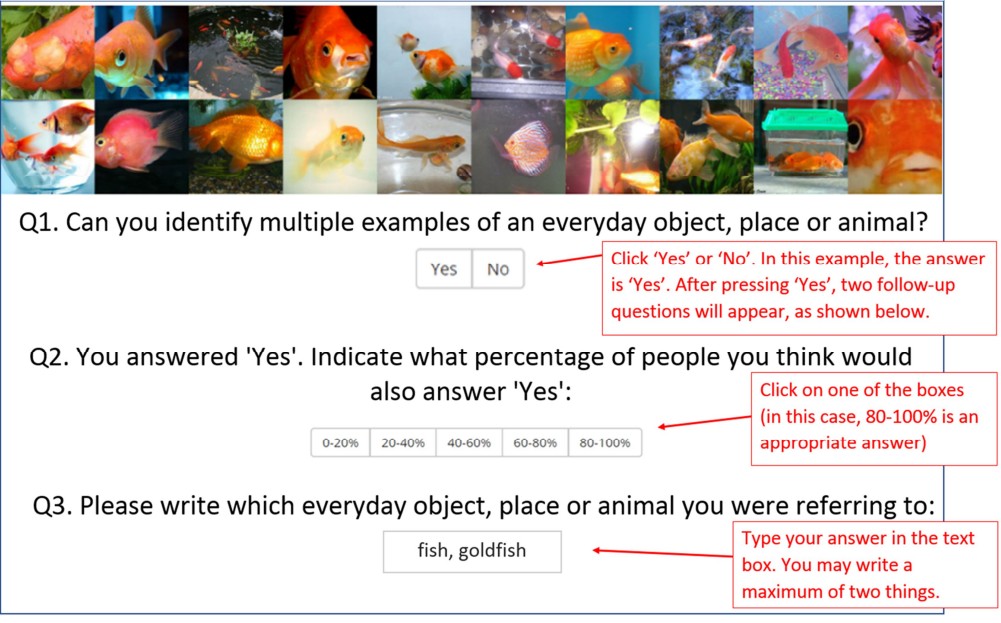

Figure A1: Example screen from the identification task shown to participants as part of the instructions. The images included on this practice trial are ImageNet2012 images, not AM images.

Some examples of the 10x10 grids of activation maximisation images that were presented to participants are shown in Figures A2, A3 and A4. Figure A2 shows an example from conv5 that human participants agreed had no obvious object in common (although there are repeated shape motifs, the participants were specifically asked for objects, and not abstract concepts like shape or color. Figure A3 is also from the conv5 and was judged by participants as some images containing 'dogs'.

Figure A4 is the AM images for the supposed 'butterfly detector' unit example discussed in the paper.

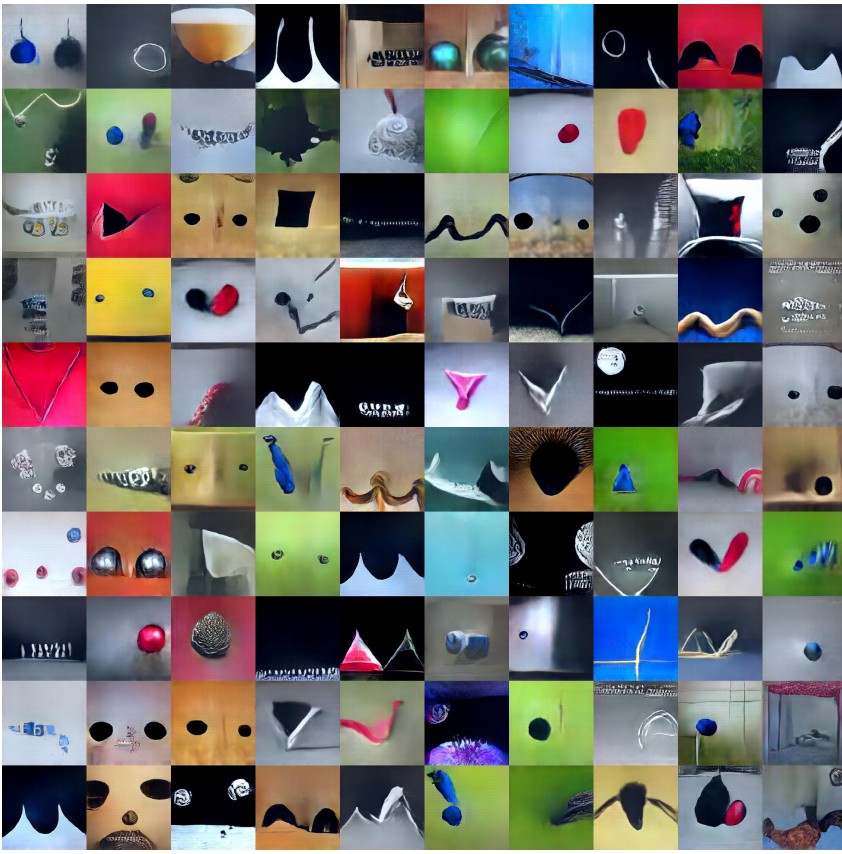

Figure A2: Example activation maximisation images for unit conv5.65. These images were judged by humans to not contain any interpretable objects in common (although the reader may agree that there are some shape and colour similarities in the images).

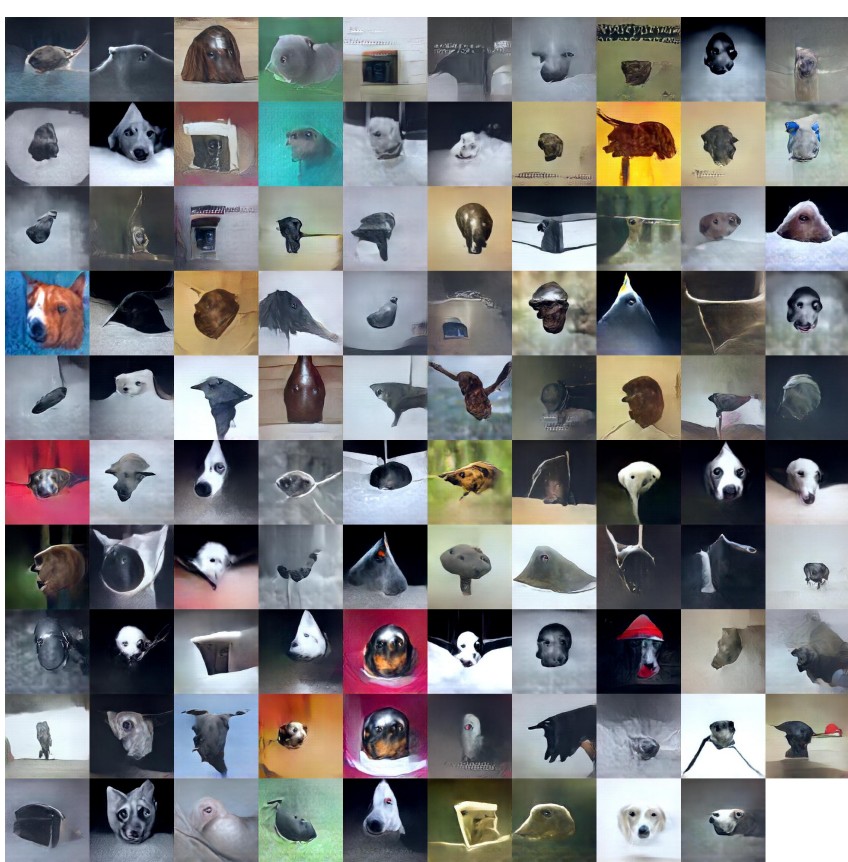

Figure A3: Example activation maximisation images for unit conv5.183. These images were judged by humans to contain some interpretable images, in this case, of the type 'dogs'.

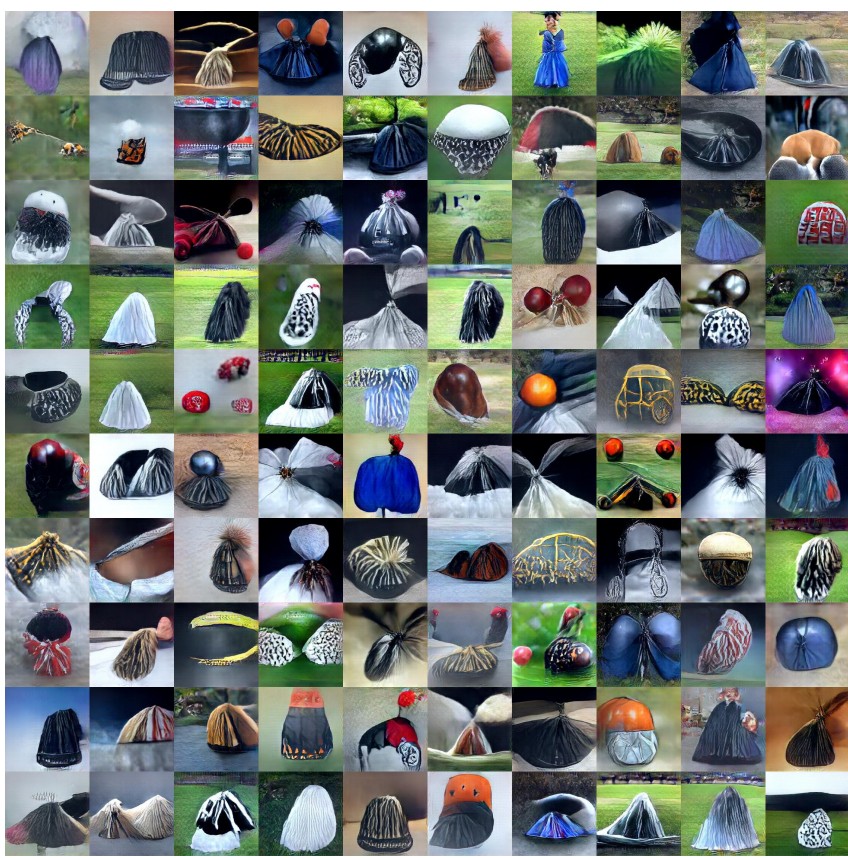

Figure A4: Example activation maximisation images for unit fc6.1199. Whilst there are some butterfly wing shapes in these images, there are not obvious butterflies. N.B. the second highest activating class for this unit is ladybirds, and there are some orange round shapes that could conceivably be ladybug-alikes.

## B    FURTHER DATA ON THE SELECTIVITY MEASURES ACROSS ALEXNET

Table  A1 gives the highest values of CCMAS and precision for each layer in AlexNet. It is worth noting that the highest CCMAS score of all hidden units was .94 (fc7.31), which at first glance suggests that this unit is close to 'perfect' selectivity. However, this unit only has low a precision score of 11%. In other words, although the mean activation for the given class is very high relative to the mean of all other activations (high CCMAS), the proportion of items from that class in the 100 most active items is low (low precision). See appendix Sec. C for discussion of how this occurs and Fig. A5(a) for an illustrative example.

| LAYER.UNIT | CCMAS | Precision |
|---|---|---|
| **Top CCMAS units** | | |
| output.322 | 0.991 | 100% |
| fc7.31 | 0.94 | 11% |
| fc6.582 | 0.93 | 1% |
| conv5.78 | 0.75 | 5% |
| **Top precision units** | | |
| output.0 | 0.99 | 100% |
| fc7.255 | 0.90 | 97% |
| fc6.1199 | 0.92 | 95% |
| conv5.0 | 0.55 | 77% |

Table A1: The units with the highest CCMAS and precision scores in AlexNet. Unit fc6.1199 was displayed in Fig. 3.

Table  A2 shows positive correlations between four of the selectivity measures used. There are moderate positive correlations between precision and CCMAS; and precision and Recall at 95% precision. The other correlations between selectivity measures have weak positive correlations. All four selectivity measures are negatively correlated with the number of classes present in the 100 most active items, that is, the more selective the unit, the fewer classes will be represented in the most active 100 items.

| | CCMAS | $recall_{0.95}$ | Max. Inf. | No. classes in top100 |
|---|---|---|---|---|
| precision | 0.38 | 0.30 | 0.15 | -0.68 |
| CCMAS | | 0.09 | 0.14 | -0.47 |
| $recall_{0.95}$ | | | 0.10 | -0.19 |
| Max. Inf. | | | | -0.22 |

Table A2: The correlations between the different measures. (All $p$'s $< .001$)

## C    FURTHER ISSUES WITH THE CCMAS MEASURE

The CCMAS measure is based on comparing the mean activation of a category with the mean activation for all other items, and this is problematic for a few reasons. First, in many units a large proportion of images do not activate a unit at all. For instance, our butterfly 'detector' unit fc6.1199 has a high proportion of images with an activation of 0.0 (see figure  3). Indeed, the inset on the middle figure shows that the distribution can be better described by exponential-derived fits rather than a Gaussian. This means that the CCMAS selectivity is heavily influenced by the the proportion of images that have an activation value of zero (or close to zero). This can lead to very different estimates of selectivity for CCMAS and precision or localist selectivity, which are driven by the most highly activated items. In  A5 we generate example data to highlight ways in which CCMAS score may be non-intuitive. In subplot (a) we demonstrate that a unit can have a CCMAS score of of 1.0 despite only a single item activating the unit. The point that we wish to emphasise is that a high CCMAS score does not necessarily imply selectivity for a given class, but might in fact relate to selectivity for a small subset of items from a given class, and this is especially true when a unit's

activation is sparse (many items do not activate the unit). However, the reverse can also be true. In subplot (c) we demonstrate that a unit can have a very low CCMAS score of .06 despite all of the most active items being from the same class.

In addition, if the CCMAS provided a good measure of a unit's class selectivity, then one should expect that a high measure of selectivity for one class would imply that the unit is not highly selective for other classes. However, the CCMAS score for the most selective category and the second most selective category (CCMAS2) were similar across the conv5, fc6 and fc7. layers, with the mean CCMAS scores .491, .844, and .848, and the CCMAS2 scores .464, .821, .831. For example, unit fc7.0 has a CCMAS of .813 for the class 'maypole', and a CCMAS2 score of .808 for 'chainsaw' (with neither of these categories corresponding 'orangutan' that had the highest precision of score of 14%).

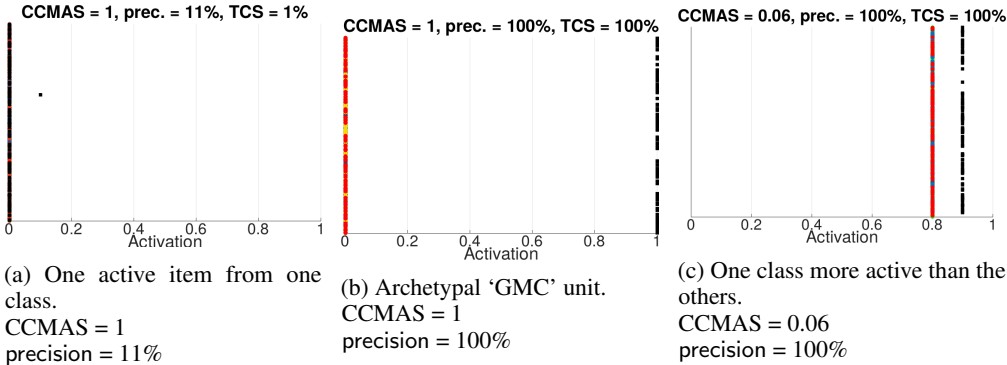

(a) One active item from one class.
CCMAS = 1
precision = 11%

(b) Archetypal 'GMC' unit.
CCMAS = 1
precision = 100%

(c) One class more active than the others.
CCMAS = 0.06
precision = 100%

Figure A5: Example of where the CCMAS does not match intuitive understandings of selectivity. Generated example data: (a) If a unit is off to all but a single image from a large class of objects, the CCMAS for that class is 1 (maximum possible selectivity). (b) An archetypal 'grandmother' cell (GMC), where the unit is strongly activated to all members of one class and off to everything else. The CCMAS is the same for (b) as for (a) although the precision is vastly different. (c): If a unit has high activations for all classes, but one class (black squares) is 0.1 more than all others (coloured circles), the CCMAS is very low (0.06) despite being %100 precision. The generated examples are for 10 classes of 100 items

## D    TESTING UNITS IN OTHER MODELS

To investigate units claimed by Zhou et al. (2018a) to be object detectors, we focus on units from a single layer that are reported to be 'bus detectors', that is, units with an IoU $\geq$ .04. We used the first 100 images per class from the ImageNet 2012 dataset as our test data. There are three classes of bus in this dataset: 'n04146614 school bus', 'n04487081 trolleybus, trolley coach, trackless trolley', 'n03769881 minibus', and this corresponded to 300 items out of 100000 images. Data for all bus unit detectors for VGG trained on places 365 are shown in table A3; for GoogLeNet trained on places 365 in table A4; and for GoogLeNet trained on ImageNet are shown in table A5. Note that for all units there are very few busses with activation at zero and that the mean activation for busses is higher than the mean activation for non-busses. However, all precision scores are all below .6, meaning that of the 100 items that most strongly activated the unit, at least 40 of them were not busses. Together these results suggests that whilst these units demonstrate some sensitivity to busses, they show poor specificity for busses (e.g., high false-alarm rate).

| unit | IoU | no.$a_x$>0 $x \in A$ | no.$a_x$>0 $x \in \neg A$ | $\mu_A$ | $\mu_{\neg A}$ | precision | CCMAS |
|---|---|---|---|---|---|---|---|
| conv5_3$_{191}$ | .15 | 99.0% | 63.9% | 131.9 | 16.1 | .45 | .78 |
| conv5_3$_{20}$ | .15 | 99.0% | 49.1% | 157.6 | 15.2 | .53 | .82 |
| conv5_3$_{333}$ | .08 | 99.0% | 71.4% | 101.7 | 17.5 | .24 | .71 |
| conv5_3$_{145}$ | .07 | 97.3% | 61.7% | 75.5 | 12.5 | .19 | .72 |
| conv5_3$_{113}$ | .06 | 97.4% | 41.0% | 62.8 | 9.1 | .12 | .75 |
| conv5_3$_{443}$ | .04 | 95.3% | 38.2% | 59.3 | 8.1 | .12 | .76 |
| conv5_3$_{131}$ | .04 | 93.7% | 22.3% | 54.0 | 5.86 | .08 | .80 |

Table A3: Selectivity measures for VGG-16, trained on Places-365, top convolutional layer units identified by Zhou et al. (2018a) as object detectors. Standard errors not shown for space, but were below $\pm 5$. The IoU is from Zhou et al. (2018a)'s network dissection method. no.$a_x$>0 and no.$a_x$>0 $x \in A$ refer to the proportion of activations that were greater than zero for busses and non-busses respectively. $\mu_A$ and $\mu_{\neg A}$ are the class means for busses and non busses respectively.

| unit | IoU | correct | no.$a_x$>0 $x \in A$ | no.$a_x$>0 $x \in \neg A$ | $\mu_A$ | $\mu_{\neg A}$ | precision | CCMAS |
|---|---|---|---|---|---|---|---|---|
| incep4e$_{824}$ | .17 | T | 100.0 | 91.4 | 41.0 | 11.8 | .27 | .55 |
| incep4e$_{745}$ | .13 | T | 98.3 | 74.8 | 34.8 | 11.4 | .06 | .51 |
| incep4e$_{791}$ | .11 | T | 98.3 | 71.4 | 32.7 | 5.3 | .41 | .72 |
| incep4e$_{194}$ | .11 | F | 100.0 | 85.3 | 26.6 | 8.8 | .02 | .51 |
| incep4e$_{82}$ | .11 | T | 100.0 | 97.3 | 26.7 | 10.9 | .14 | .42 |
| incep4e$_{736}$ | .11 | T | 100.0 | 78.8 | 38.7 | 9.9 | .05 | .59 |
| incep4e$_{663}$ | .10 | F | 96.0 | 38.0 | 33.4 | 3.7 | .15 | .80 |
| incep4e$_{94}$ | .10 | T | 100.0 | 91.6 | 38.3 | 9.5 | .35 | .60 |
| incep4e$_{772}$ | .08 | F | 97.3 | 54.6 | 21.7 | 5.2 | .00 | .61 |
| incep4e$_{113}$ | .08 | F | 100.0 | 88.0 | 24.9 | 9.2 | .02 | .46 |
| incep4e$_{708}$ | .06 | F | 100.0 | 85.1 | 29.7 | 9.1 | .02 | .53 |
| incep4e$_{801}$ | .06 | F | 100.0 | 64.5 | 35.2 | 6.4 | .14 | .69 |
| incep4e$_{199}$ | .06 | F | 99.7 | 92.2 | 21.5 | 7.7 | .09 | .47 |
| incep4e$_{8}$ | .05 | F | 99.7 | 83.5 | 18.5 | 7.3 | .01 | .43 |
| incep4e$_{121}$ | .05 | F | 100.0 | 90.4 | 17.9 | 8.9 | .01 | .34 |
| incep4e$_{622}$ | .05 | T | 96.0 | 65.0 | 27.5 | 6.4 | .20 | .62 |
| incep4e$_{97}$ | .04 | T | 99.3 | 86.4 | 21.1 | 9.3 | .04 | .39 |

Table A4: Selectivity measures for GoogLeNet, trained on Places-365, layer inception4e units identified by Zhou et al. (2018a) as object detectors. Standard errors not shown for space, but were below $\pm 2$. The IoU is from Zhou et al. (2018a)'s network dissection method. A units is marked as correct if there was a single bus in the 4 example pictures on the website (http://netdissect.csail.mit.edu/dissect/googlenet_places365/), and false if not. Where units are False, this might suggest that the units were responding to 'bus like' features in none bus objects. no.$a_x$>0 and no.$a_x$>0 $x \in A$ refer to the proportion of activations that were greater than zero for busses and non-busses respectively. $\mu_A$ and $\mu_{\neg A}$ are the class means for busses and non busses respectively.

| unit | IoU | correct | no.$a_x$>0 $x \in A$ | no.$a_x$>0 $x \in \neg A$ | $\mu_A$ | $\mu_{\neg A}$ | precision | CCMAS |
|---|---|---|---|---|---|---|---|---|
| incep4e$_{494}$ | .11 | F | 99.0 | 82.4 | 72.5 | 22.8 | .00 | .52 |
| incep4e$_{828}$ | .10 | T | 100.0 | 72.6 | 109.4 | 17.6 | .45 | .72 |
| incep4e$_{569}$ | .10 | T | 99.7 | 85.9 | 74.9 | 20.0 | .05 | .58 |
| incep4e$_{384}$ | .10 | T | 100.0 | 71.6 | 67.0 | 18.5 | .00 | .57 |
| incep4e$_{455}$ | .09 | T | 99.7 | 89.6 | 69.1 | 14.3 | .3 | .66 |
| incep4e$_{579}$ | .09 | T | 100.0 | 97.0 | 91.5 | 26.0 | .23 | .56 |
| incep4e$_{331}$ | .08 | T | 98.0 | 75.5 | 51.0 | 11.8 | .12 | .62 |
| incep4e$_{582}$ | .08 | T | 100.0 | 83.4 | 125.7 | 21.95 | .58 | .70 |
| incep4e$_{498}$ | .07 | T | 97.7 | 77.2 | 73.5 | 15.0 | .52 | .66 |
| incep4e$_{534}$ | .07 | F | 99.3 | 81.2 | 62.7 | 19. | .02 | .53 |
| incep4e$_{693}$ | .07 | T | 98.7 | 91.2 | 75.4 | 22.3 | .15 | .54 |
| incep4e$_{673}$ | .07 | T | 99.7 | 88.4 | 88.6 | 23.0 | .33 | .59 |
| incep4e$_{469}$ | .06 | T | 98.7 | 78.1 | 34.7 | 14.6 | .00 | .41 |
| incep4e$_{207}$ | .06 | T | 100.0 | 93.5 | 76.1 | 21.3 | .07 | .56 |
| incep4e$_{491}$ | .06 | F | 99.0 | 74.5 | 41.1 | 13.7 | .01 | .50 |
| incep4e$_{645}$ | .06 | T | 98.0 | 83.9 | 59.9 | 18.1 | .20 | .54 |
| incep4e$_{527}$ | .06 | F | 100.0 | 91.5 | 58.0 | 21.7 | .00 | .46 |
| incep4e$_{511}$ | .05 | F | 100.0 | 89.4 | 53.5 | 21.7 | .00 | .42 |
| incep4e$_{308}$ | .05 | F | 100.0 | 89.4 | 53.5 | 21.7 | .00 | .42 |
| incep4e$_{541}$ | .05 | F | 99.67 | 88.7 | 44.9 | 13.7 | .00 | .53 |
| incep4e$_{367}$ | .05 | T | 97.3 | 80.3 | 37.7 | 15.4 | .02 | .42 |
| incep4e$_{665}$ | .05 | T | 100.0 | 82.45 | 107.2 | 21.0 | .33 | .67 |
| incep4e$_{532}$ | .05 | T | 100.0 | 91.5 | 52.9 | 22.4 | .05 | .41 |
| incep4e$_{297}$ | .04 | T | 99.7 | 90.2 | 48.2 | 17.9 | .00 | .46 |
| incep4e$_{480}$ | .04 | T | 100.0 | 92.9 | 69.4 | 21.4 | .02 | .53 |

Table A5: Selectivity measures for GoogLeNet, trained on ImageNet, layer inception4e units units identified by Zhou et al. (2018a) as object detectors. Standard errors not shown for space, but were below $\pm2$. A units is marked as correct if there was a single bus in the 4 example pictures on the website (http://netdissect.csail.mit.edu/dissect/googlenet_imagenet/), and false if not. This might suggest that the units were responding to 'bus like' features in none bus objects.

