# OpenReview forum: "Are there any 'object detectors' in the hidden layers of CNNs trained to identify objects or scenes?"
_ICLR.cc/2020/Conference — Reject_

### Official Review · AnonReviewer3 · 2019-10-15
**Official Blind Review #3**

**Rating:** 3

**Review:**

The paper makes an empirical claim that CNNs for object recognition do not contain hidden neuron which is highly selective to each class, mainly based on three aspects: (a) metrics related to the maximum informedness, (b) jitterplots of activation data, and (c) a user study assessing whether generated images maximizing a given unit is perceptible to the user. The paper point out these results are in contrast to the case of RNN, where it has been reported that many localist hidden units emerge. It is also noticed that the existing metrics for selectivity do not adequately discriminate highly selective units in CNN.

In overall, the manuscript is well-written and easy-to-follow. I particularly appreciated a kindly presented overview on the literature and thoroughly conducted experiments including a complete user study. One of my key concerns, however, is that I am still not fully convinced whether the key finding in this paper - the lack of highly selective units in CNNs - is an indeed important problem for ICLR community: Personally, I feel the "existence" of selective units in RNN could be interesting, but the "non-existence" in the case of CNN is not that surprising for some readers, as it seems much likely (at least to me): The final layer of CNN would be surely selective across classes, but it may be not the case for the hidden layers - Nevertheless, some of the units may act selectively, not across classes but in some other concepts: e.g. stripes, orientations, etc. Therefore, I wonder if the paper could further provide a discussion on the importance of the key finding.

- In Section 2 - Network and Dataset - "... We selected 233 ..." : Which criteria is actually used to choose the candidate units for the analysis?
- Do the overall results mean that the "maximum informedness"-based metrics are superior to the others for assessing selectivity of a unit? Also, are those metrics original to this paper?
- Currently, I feel the concept of "selectivity" is presented in somewhat subjectively: the definition could vary across the context. It would be nice if this could be more formalized to support the claims in the paper, e.g. the superiority of the proposed metrics.

**Experience Assessment:**

I do not know much about this area.

**Review Assessment: Checking Correctness Of Derivations And Theory:**

N/A

**Review Assessment: Checking Correctness Of Experiments:**

I carefully checked the experiments.

**Review Assessment: Thoroughness In Paper Reading:**

I read the paper at least twice and used my best judgement in assessing the paper.

---

> ### Author Response · Authors · 2019-11-11
> **Response to Reviewer 3**
>
> REVIEWER 3 WRITES: In overall, the manuscript is well-written and easy-to-follow. I particularly appreciated a kindly presented overview on the literature and thoroughly conducted experiments including a complete user study. One of my key concerns, however, is that I am still not fully convinced whether the key finding in this paper - the lack of highly selective units in CNNs - is an indeed important problem for ICLR community…Therefore, I wonder if the paper could further provide a discussion on the importance of the key finding.
> RESPONSE: We have tried to make clearer in our responses above (and in revision of our manuscript) the theoretical significance of our findings.  Most importantly, the different selectivity measures give very different impressions of selectivity, and this has led to researchers to use similar terms to describe the selective units in CNNs (object detectors) and recurrent neural networks (localist representations and grandmother cells). This obscures the fact that the level of selectivity in these networks can be dramatically different (maximum recall with perfect selectivity in CNNS is 8%, maximum with recurrent networks is 100%).  More work needs to be done to assess the level of selectivity across different networks to better understand the role of the superposition catastrophe and other factors in determining degree of selectivity, and our findings provide the tools needed to carry out this work.
>
> REVIWER 3 WRITES:  In Section 2 - Network and Dataset - "... We selected 233 ..." : Which criteria is actually used to choose the candidate units for the analysis?
>
> RESPONSE:  These were just randomly selected images and now note this in the paper.
>
> REVIWER 3 WRITES:  Do the overall results mean that the "maximum informedness"-based metrics are superior to the others for assessing selectivity of a unit? Also, are those metrics original to this paper?
>
> RESPONSE: The use of maximum informedness is a standard signal detection measure and is the first it has been applied to neural networks. We think it is better than CCMAS and precision for a number of reasons. First, the CCMAS and precision provide very high selectivity scores that do not intuitively match up with the actual level of selectivity that can be directly viewed in the jitterplots we report.  Second, as we now highlight in the Appendix C, CCMAS have a number of additional problems that make the measure misleading and unsuitable, namely; (1) high selectivity for one class does not imply that is also highly selective for another class, and (2) the same highly selective CCMAS score is consistent with a wide range of outcomes, including cases in which the unit is clearly not selective for a category.  But the most important point is not that our signal detection measures are better (although we think they are).  As Reviewer 2 notes, it is a good thing to have multiple measures to better capture the complex data summarized in scatter plots.  Rather, it is that different measures provide very different estimates, and the failure to appreciate this can (and has) led to confusions.
>
> REVIWER 3 WRITES:  - Currently, I feel the concept of "selectivity" is presented in somewhat subjectively: the definition could vary across the context. It would be nice if this could be more formalized to support the claims in the paper, e.g. the superiority of the proposed metrics.
>
> RESPONSE:  As discussed above, it is appropriate to have multiple selectivity measures (although the CCMAS measure is problematic) as they can highlight different aspects of the selectivity.  Giving a formal definition by focusing on one measure might be counterproductive and unnecessarily controversial.  But it is important to use signal detection measures that better characterize our intuitive sense of selectivity compared to CMMAS, precision, and network dissection measures (e.g., the high CCMAS and precision scores seem at odds with the selectivity as shown in the jitterplots).  It is also important to use common selectivity measures across papers and models in order to avoid conceptual confusions.  We try and make this main message more clearly in the current version.

---

### Official Review · AnonReviewer2 · 2019-10-17
**Official Blind Review #2**

**Rating:** 3

**Review:**

The paper empirically studies the category selectivity of individual cells in hidden units of CNNs. It is a sort of "meta-study" and comparison of different metrics proposed to identify cells with a preference for a specific target category. The claimed finding is that there are no cells that are "sufficiently" selective to be called object detectors.

The paper is seemingly motivated by the authors' perceiving a contradiction: it is assumed that the power of neural networks is (among others) due to the distributed representation; whereas the presence of object detectors would, in the extreme case, mean that the representation is disentangled into a separate unit per category. It may be a matter of terminology, but this is where my disagreement with the authors start. I do not see a simplistic dichotomy, where one could or should determine which of the two interpretations is "right" or "wrong". In my view, which I believe is the mainstream interpretation, a distributed representation does not contradict the presence of specialised units. Some categories probably are easily identified by few distinctive features, so there will be more detector-like units; others are complex and hence more diffusely spread through the network; and of course there is no guarantee that the learned "object detectors" are tuned to exactly the target categories, after all it is the purpose of the network to gradually translate the data distribution to the label distribution - if the categories were directly apparent in the data, nearest-neighbour would be enough. So it is not only possible, but rather likely that the learned "object detectors" are to some degree driven by the statistics of the data, not the labels - e.g., there could be a highly selective "bird" unit which nevertheless has high false positive rate for any of the more specific bird species categories in the imageNet nomenclature. And vice versa, there could be a highly specific "Ferrari" detector that is so specialised that it has low recall for the "sports car" class (this case includes, among others, the case of viewpoint-specific detectors for certain categories). In the words of the paper, the "selective units are sensitive to some feature that is frequently, but not exclusively associated with the class" - I thought this is the standard majority view, not a surprising finding. In this context terminology matters: the study effectively tries to disprove that the network learns "near-perfect single output-category detectors", but who claims that it would do that?

I agree with the authors that there is by now a zoo of selectivity metrics that are not always highly correlated. But is that a problem? We have a zoo of quality metrics for many machine learning problems - that is not necessarily a weakness, but simply reflects the obvious fact that a single number is not enough to characterise performance in a complex cognitive task. It is the job of the researcher/user to chose the metric that s most suitable for their specific question, and to correctly interpret its numerical value.

Regarding the methodology, the paper did a lot of work to systematically crunch the numbers and analyse network units. It is a laudable effort that someone took on that job. A few technical decisions are unclear to me. Why analyse only some of the units? If one collects statistics over >2000 units of a fully connected layer, one might as well do the complete job and use all 4096 units. Similarly, why analyse only the correctly classified images? While it is clear that one must separately look at them, also the activations on incorrectly classified ones could provide valuable insights. E.g., do false positives of class X on average activate a certain "class X detector"? Why chose only the class with highest mean activation for CCMAS? That might be unrepresentative, e.g., a neuron might, for that particular class, always have high activation due to some very common background context, and still be not selective at all.

Regarding the results, I find them much less clear-cut than the paper claims. For example, I find it quite remarkable that some unit has 8% recall at perfect precision. After all, only approximately 0.1% of the images are in the correct category, so a unit that flags 8% of them without making a mistake is a pretty good detector for (part of) the target class, cf Fig. 3. Also regarding Fig. 2 / maximum informedness, the statistics actually do not look bad. Of course false alarm proportion remains high - but the chance level here is 99.9%, so even a 99% false alarm rate means that your unit can, on its own, reject 90% of the true negatives. I find the proposed "minimum condition" for an object detector (>50% recall at >50% precision) unrealistic: the top-1 accuracy of AlexNet is, to my knowledge, <63%. Even the complete network probably never reaches 50% recall for most classes.

Especially the user study - which is again a commendable effort - in my view does not confirm the claims. According to that study, almost 60% 0f all fc8 units are "object detectors", with very high conherence between humans and selectivity metrics.

Overall, while it is an interesting study, it remains unclear to me what I should learn from it. I don't see why different measures provide "misleading conclusions" that need to be rectified. Conclusions are the responsibility of the researcher interpreting the numbers, not of the formula to calculate some statistical performance metric. I am in a difficult situation here: the study is one of those things (like determining human performance on ImageNet, or re-coding some baseline where the original code is not available) where I find it valuable that someone did them in the community, but still I don't think they need a reviewed paper.  A note on the blog, or on arXiv, is enough.


**Experience Assessment:**

I have read many papers in this area.

**Review Assessment: Checking Correctness Of Derivations And Theory:**

I assessed the sensibility of the derivations and theory.

**Review Assessment: Checking Correctness Of Experiments:**

I assessed the sensibility of the experiments.

**Review Assessment: Thoroughness In Paper Reading:**

I read the paper at least twice and used my best judgement in assessing the paper.

---

> ### Author Response · Authors · 2019-11-11
> **Response to Reviewer 2**
>
> REVIEWER 2 “ WRITES: “In my view, which I believe is the mainstream interpretation, a distributed representation does not contradict the presence of specialised units…”.
>
> **RESPONSE:  We agree that distributed and selective codes co-occur.  But this is different than the question we are asking, namely, how selective are the most selective units in CNNs involved in object identification?  Note, the Reviewer’s examples of high false alarm rates to “birds” and low recall rates to “sports car” are both somewhat misleading.  We found high false alarm rates to all categories in ImageNet (not only categories with many subordinate ones like the category “birds”), and we computed recall rates on the specific trained categories (in this example “Ferrari”) not untrained superordinate categories (“sports car”).   Indeed, the most selective unit we found in ImageNet was Monarch Butterfly, that according to this logic, this unit should shower higher false alarms and lower recall scores.
>
> Regarding whether anyone predicts perfect selectivity, it has already been demonstrated that recurrent networks sometimes learn 100% selective units (Bowers 2014, 2016), although this work was published in psychology journals that researchers in computer science are likely unfamiliar with.  And although not claiming perfect selectivity, Zhou et al. reports “object detectors” with precision scores of >90% and Morcos et al. report CCMAS scores approaching 1.0  But when you apply the signal detection measures you find that the most selective units have false alarm rates of ~99% and maximum recall rates of < 8% with perfect precision.
>
> REVIEWER 2 WRITES: “I agree with the authors that there is by now a zoo of selectivity metrics that are not always highly correlated. But is that a problem?”
>
> **RESPONSE: We are not claiming that only one metric can be used, but this is the first demonstration that the different metrics often produce very different estimates of selectivity.  As noted above, this can lead to confusions: e.g., the terms “object detector” in response to high precision measures in CNNs and “localist unit” in response to high signal detection measures in recurrent networks seem to suggest similar types of representations are learned.  This is not the case.
>
> REVIEWER 2 WRITES: "A few technical decisions are unclear to me... "
>
> RESPONSE:  We sampled a large proportion of units in AlexNet and there is no reason to expect differences in the other units.  Consistent with this, when we probed the most selective units in other CNNs as identified by Zhou et al. we again found that the units were only moderately selective, and no way can be reasonably characterized as “object detectors”.  We focused on correctly identified images simply because the question we are asking, namely, do CNNs learn object detectors in order to correctly identifying objects?  With regards to the CCMAS measure, we only focused on the most category with the highest CCMAS score (as did Morcos et al.).  However, in response to this point, we now note in the end of Appendix C that the CCMAS measure of the second most active category is similar to the CCMAS measure of the most active category.  This is problematic: If the CCMAS provided a good measure of a unit’s class selectivity then one should expect that a high measure of selectivity for one class would imply that the unit is not highly selective for other classes.
>
> Note, there is another important problem with the CCMAS measure that we previously detailed in the Appendix (see Figure 12 in the supplement from reviewed version), now Figure A5 in the Appendix C, namely, that the same CCMAS measure can reflect very different activations as reflected in the jitterplots, some of which are not selective at all.  This highlights how this measure is not fit for purpose in measuring selectivity in NNs (we now refer directly to this information in the Appendix in the body of the paper, see page 8).
>
> REVIEWER 2: Regarding the results, I find them much less clear-cut than the paper claims..."
>
> RESPONSE:  We do not think there is anything unclear with our results, but we are giving different interpretations to our signal detection outcomes.   The author might find the 8% recall with perfect precision remarkable, but this level of selectivity gives a very different impression than reports of object detectors with over 75% precision, and it is very different than the grandmother cells reported in recurrent networks that have 100% recall with perfect precision.  Our selectivity measures were based on images that were correctly identified with top-1 accuracy.  We agree that there is no objective criterion for “object detector”, but we don’t expect describing a unit as an “object detector” with > 75% precision leads readers to infer that the same  unit has a false alarm rate of nearly 99% or very low hit rate.  We highlight the subjective nature of the term more clearly in the current version of article.

---

### Official Review · AnonReviewer1 · 2019-10-22
**Official Blind Review #1**

**Rating:** 8

**Review:**

This work investigates the collection of methods that have been proposed to find units in neural networks that are selective for certain object classes.  Previous works have used different measures of selectivity (with sometimes contradictory results), and the authors investigate the degree to which these units qualify as “object detectors”.

This research area is important for understanding deep networks because claims have been made as to the relative importance (or lack thereof) of these individual units as identified by different measures vis-a-vis distributed representations -- the identification of such units would be interesting for understanding the predictions of classification networks.

The authors find that (1) different proposed measures of selectivity are not consistent and (2) units identified as selective cannot be considered object detectors due to the high false alarm / low hit rates, analyzing a large number of selectivity measures.  I would have liked to see experiments on more recent architectures (the focus of the paper is on a dated architecture (AlexNet)); there is analysis on units in GoogLeNet and VGG-16 but it would also be interesting to see results for more modern architectures (e.g. DenseNet and ResNet).

Overall, I think that the authors have presented a strong meta-analysis and compelling argument for further study in rigorously identifying the presence (or lack thereof) of selective units in neural networks and the degree to which they may be considered "object detectors."

**Experience Assessment:**

I do not know much about this area.

**Review Assessment: Checking Correctness Of Derivations And Theory:**

N/A

**Review Assessment: Checking Correctness Of Experiments:**

I assessed the sensibility of the experiments.

**Review Assessment: Thoroughness In Paper Reading:**

I read the paper at least twice and used my best judgement in assessing the paper.

---

### Author Response · Authors · 2019-11-11
**GENERAL RESPONSE**

We are pleased that Reviewer 1 was so positive, but of course disappointed that Reviewers 2-3 did not find our results sufficiently important/surprising to recommend higher ratings.  Reviewer 2 summarize his/her thoughts as: “Overall, while it is an interesting study, it remains unclear to me what I should learn from it. I don't see why different measures provide "misleading conclusions" that need to be rectified”, and Reviewer 3 writes: “One of my key concerns, however, is that I am still not fully convinced whether the key finding in this paper - the lack of highly selective units in CNNs - is an indeed important problem for ICLR community: Personally, I feel the "existence" of selective units in RNN could be interesting, but the "non-existence" in the case of CNN is not that surprising for some readers”.

We respond in more detail to their concerns below, but here we wanted to just briefly highlight why these results are both empirically and theoretically important.  First, ICLR has published several papers reporting highly selective representations in NNs, including:

D. Bau, J. Zhu, H. Strobelt, B. Zhou, J. B. Tenenbaum, W. T. Freeman, A. Torralba. (2019) GAN Dissection: Visualizing and Understanding Generative Adversarial Networks. ICLR. (cited 33 according to Google Scholar)

Morcos, A. S., Barrett, D. G., Rabinowitz, N. C., & Botvinick, M. (2018). On the importance of single directions for generalization. ICLR. (cited 74 times)

Karpathy, A., Johnson, J., & Fei-Fei, L. (2016). Visualizing and understanding recurrent networks. ICLR, (cited > 600 times).
Zhou, B., Khosla, A., Lapedriza, A., Oliva, A., & Torralba, A. (2015). Object detectors emerge in deep scene cnns. ICLR (cited >330)

Our paper clarifies the level of selectivity that is observed in single units in CNNs, and it provides the first systematic comparison of different measures of selectivity (including the human interpretation of activation maximization images).  We show that units with CCMAS measures of near > .95 and precision measures of >.9, are only moderately selective accordingly signal detection measures.

Second, we show that the use of different selectivity measures can lead to confusions. For example, describing single units as “object detector” in response to high precision measures (Zhou et al.) and “localist units” in response to high signal detection measures (Bowers et al., 2014) might suggest similar types of representations are learned in CNNs and recurrent networks (RNNs), respectively.  Indeed, we are not aware of anyone in the machine learning community who has even considered the hypothesis that selectivity is reduced in CNNs compared RNNs.  But here we show that the moderate levels of selectivity found in various CNNs contrasts with the highly selective (sometimes 100% selective grandmother cell) codes reported in recurrent NNs (e.g., Bowers et al., 2014, 2016; Karpathy et al., 2016; Lakretz, Lakretz et al., 2019).  Bowers et al., (2014) argued that more selective codes are observed in recurrent networks when they learn to overcome the superposition constraint, and this hypothesis is supported by the current results (a hypothesis that has not made in ICLR or any computer science journals).  We highlight this theoretical point more strongly in the current version of the article in the General Discussion.  In sum, we think our use of signal detection measures to characterize single-unit selectivity in CNNs provides an important empirical and theoretical contributions.

Lakretz, Y., Kruszewski, G., Desbordes, T., Hupkes, D., Dehaene, S., & Baroni, M. (2019). The emergence of number and syntax units in LSTM language models. arXiv preprint arXiv:1903.07435.

---

### Decision · Program_Chairs · 2019-12-19

**Decision:**

Reject

**Comment:**

This paper conducted a number of empirical studies to find whether units in object-classification CNN can be used as object detectors. The claimed conclusion is that there are no units that are sufficient powerful to be considered as object detectors. Three reviewers have split reviews. While reviewer #1 is positive about this work, the review is quite brief. In contrast, Reviewer #2 and #3 both rate weak reject, with similar major concerns. That is, the conclusion seems non-conclusive and not surprising as well. What would be the contribution of this type of conclusion to the ICLR community? In particular, Reviewer #2 provided detailed and well elaborated comments. The authors made efforts to response to all reviewers’ comments. However, the major concerns remain, and the rating were not changed. The ACs concur the major concerns and agree that the paper can not be accepted at its current state.